# Beta-Sitosterol Enhances Classical Swine Fever Virus Infection: Insights from RNA-Seq Analysis

**DOI:** 10.3390/v17070933

**Published:** 2025-06-30

**Authors:** Yayun Liu, Dongdong Yin, Jieru Wang, Yin Dai, Xuehuai Shen, Lei Yin, Bin Zhou, Xiaocheng Pan

**Affiliations:** 1Institute of Animal Husbandry and Veterinary Science, Anhui Academy of Agricultural Sciences, Livestock and Poultry Epidemic Diseases Research Center of Anhui Province, Hefei 230031, China; liuyy20210120@163.com (Y.L.); yindd160@163.com (D.Y.); wangjr0317@126.com (J.W.); dalin2080@aaas.org.cn (Y.D.); xuehuaishen1986@126.com (X.S.); yinlei1989@yeah.net (L.Y.); 2Anhui Provincial Key Laboratory of Livestock and Poultry Product Safety, Hefei 230031, China; 3MOE Joint International Research Laboratory of Animal Health and Food Safety, College of Veterinary Medicine, Nanjing Agricultural University, Nanjing 210095, China

**Keywords:** beta-sitosterol (BS), classical swine fever virus (CSFV), lipid compounds, replication, differentially expressed genes (DEGs)

## Abstract

Beta-sitosterol (BS), a naturally occurring phytosterol abundant in plants, has been reported to exhibit diverse biological activities, including immunomodulatory and antiviral effects. Classical swine fever virus (CSFV), a member of the Pestivirus genus, remains a persistent threat to the swine industry worldwide, causing considerable economic damage. Our research found that BS significantly enhances the replication of both the CSFV-Shimen strain and the attenuated C-strain vaccine virus in PK-15 cells. Additionally, transcriptomic profiling (RNA-Seq) identified 175 differentially expressed genes (DEGs) following BS exposure, comprising 53 upregulated and 122 downregulated genes. Further results demonstrated that treatment with β-sitosterol suppressed IκBα expression, thereby activating the NF-κB pathway, and that knockdown of endogenous IκBα significantly promoted CSFV replication. These findings contribute to a deeper understanding of how BS influences the CSFV infection process, suggesting its role as a host lipid-associated factor facilitating viral propagation.

## 1. Introduction

Classical swine fever virus (CSFV), classified under the genus Pestivirus in the family Flaviviridae, causes classical swine fever (CSF), a highly contagious disease responsible for severe economic damage to the global swine industry [1]. In China, vaccination remains the primary measure for controlling CSFV infection. The live attenuated C-strain vaccine has effectively prevented widespread outbreaks. However, sporadic and endemic cases continue to emerge in various regions, posing ongoing challenges to swine production and food security [2,3]. Therefore, understanding the molecular basis of CSFV infection and its interplay with host cellular machinery is crucial for advancing more efficient strategies for disease prevention and control.

Sterols are a class of steroidal compounds widely distributed across living organisms and serve as essential components of cellular membranes. Based on their origin, sterols are classified as zoosterols (animal-derived), mycosterols (fungal-derived), and phytosterols (plant-derived) [4]. Cholesterol, the predominant sterol in animal cells, is essential for maintaining membrane integrity and cellular homeostasis. In mammalian and insect models, cholesterol has been shown to be indispensable for various stages of the flavivirus life cycle. Specifically, cholesterol regulates viral entry, replication complex formation, assembly, and release during infection with dengue virus (DENV), Zika virus (ZIKV), yellow fever virus (YFV), and West Nile virus (WNV) [5]. Among phytosterols, beta-sitosterol (BS) and its glycosides (BSG) are the most abundant, exhibiting anti-inflammatory, antioxidant, and immunomodulatory activities in vitro and in vivo [6,7]. Due to its promising antiviral properties, BS is currently under investigation as a potential candidate for antiviral drug development. In influenza A virus (IAV)-infected cells, BS suppresses the inflammatory response in a dose-dependent manner by inhibiting NF-κB and p38 mitogen-activated protein kinase (MAPK) signaling, which is accompanied by reduced induction of type I and III interferons (IFNs) [8]. In silico studies have further identified BS as a potential inhibitor of hepatitis C virus (HCV), underscoring its relevance in discovering natural antiviral compounds [9].

Lipid metabolism is critical in multiple stages of the CSFV life cycle, encompassing viral attachment, cellular entry, uncoating, genome replication, assembly, and release [10]. For instance, U18666A, a compound that disrupts intracellular cholesterol transport, induces cholesterol accumulation in lysosomes and inhibits CSFV uncoating [11]. The CSFV nonstructural protein NS4B upregulates fatty acid synthase (FASN) and recruits it to the viral replication complex (VRC), supporting the fatty acid synthesis necessary for viral replication [12]. In addition, the NS5A protein hijacks the de novo sphingolipid biosynthesis pathway to facilitate virion assembly and maturation [13]. Despite these advances, the molecular mechanisms by which distinct lipid species regulate CSFV infection in PK-15 cells remain poorly understood. In the present study, BS was identified as a potent enhancer of CSFV replication. Screening of 135 lipid compounds revealed that BS significantly promoted the replication of both the CSFV-Shimen strain and the attenuated C-strain. Transcriptomic analysis demonstrated that genes differentially expressed following BS treatment were primarily involved in metabolic processes, immune defense, biological regulation, and signal transduction. These findings provide new insights into the lipid-mediated regulation of CSFV infection and establish a foundation for developing antiviral strategies targeting host lipid metabolic pathways.

## 2. Materials and Methods

### 2.1. Viruses, Cell Culture, and Lipid Compound

The CSFV-Shimen strain (GenBank ID: AF092448) and the Japanese encephalitis virus (JEV) NJ2008 strain (GenBank ID: GQ918133) were used in this study as previously documented [12]. Porcine kidney epithelial cells (PK-15) and baby hamster kidney cells (BHK-21) were cultured in Dulbecco’s Modified Eagle Medium (DMEM; GIBCO, New York, NY, USA) enriched with 10% fetal bovine serum (FBS; GIBCO, New York, NY, USA), 0.2% sodium bicarbonate (NaHCO_3_), and antibiotics (100 μg/mL streptomycin and 100 IU/mL penicillin). Cells were maintained at 37 °C in a humidified incubator with 5% CO_2_. A lipid compound library (HY-L043, MedChemExpress (MCE), Shanghai, China) was employed for compound screening [13].

### 2.2. Immunofluorescence Assay (IFA)

PK-15 cells were seeded onto culture dishes and infected with CSFV, followed by BS or vehicle control treatment. After treatment, the cells were rinsed with PBS, fixed with 4% paraformaldehyde, and permeabilized with 0.1% Triton X-100. CSFV was detected using a mouse monoclonal anti-E2 antibody (WH303), and fluorescence microscopy was used for visualization. Nuclei were counterstained with DAPI.

### 2.3. Cell Viability Assay

Cell viability was evaluated utilizing the Cell Counting Kit-8 (CCK-8) assay. PK-15 cells were seeded into 96-well plates, mock-infected or infected with CSFV, and exposed to varying concentrations of lipid compounds for 24 h. Viability was assessed according to the manufacturer’s instructions and previously established protocols [14].

### 2.4. Total RNA Extraction and RT-qPCR

The infected cells were lysed by two freeze–thaw cycles and total RNA was extracted using TRIzol reagent (TaKaRa Bio, Dalian, China). RNA copies of CSFV and cellular factors were quantified by RT-qPCR in a 7300 real-time PCR system (Applied Biosystems, Foster City, CA, USA) using a series of the specific primers as follows: CSFV forward (5′-3′): CCTGAGGACCAAACACATGTTG, CSFV reverse (5′-3′): TGGTGGAAGTTGGTTGTGTCTG; β-actin forward (5′-3′): CTCCATCATGAAGTGCGACGT, β-actin reverse (5′-3′): GTGATCTCCTTCTGCATCCTGTC. Data analysis is expressed as the 2^−ΔΔCT^ value from quadruplicate samples [15].

### 2.5. Western Blotting Analysis

Cells were washed thrice with cold PBS and lysed in cold RIPA buffer for 20 min. Lysates were clarified by centrifugation at 14,000× *g* for 10 min at 4 °C. Proteins were separated by SDS-PAGE and transferred onto nitrocellulose membranes. Immunoblotting was performed using specific primary antibodies, with β-actin serving as the internal reference for normalization. The intensity of immunoreactive bands was quantified utilizing ImageJ software (version 7.0), and protein expression levels were normalized to β-actin.

### 2.6. RNA Sequencing (RNA-Seq) Sample Preparation

PK-15 cells were seeded in 60 mm culture dishes and pretreated with 100 nM BS or an equivalent volume of DMSO for 1 h, followed by infection with CSFV. After 1.5 h, the medium was replaced with a maintenance medium containing BS, and cells were incubated for an additional 24 h. Total RNA was extracted using TRIzol reagent, and cell lysates were snap-frozen in liquid nitrogen for subsequent RNA-Seq analysis.

### 2.7. siRNA Transfections

Cells were transfected with 30 nM siRNA using siRNA-mate plus (GenePharma, Shanghai, China) according to the manufacturer’s instructions. SiRNA duplexes targeting IκBα and a negative control were synthesized by GenePharma. At 24 to 36 h post-transfection, cells were infected with CSFV for RT-qPCR and Western blotting [12].

### 2.8. Statistical Analysis

All quantitative data are presented as mean values ± standard deviation (SD). Statistical differences between experimental and control groups were assessed utilizing an unpaired two-tailed Student’s *t*-test. A *p* value less than 0.05 was considered indicative of statistical significance (*), while values below 0.01 were regarded as highly significant (**). Statistical processing and graphical representation were conducted utilizing GraphPad Prism version 6 (GraphPad Software Inc., La Jolla, CA, USA).

## 3. Results

### 3.1. Lipid Compounds Promote CSFV Proliferation

To evaluate the effects of lipid compounds on CSFV replication, an initial screening was performed using 135 lipid molecules in an immunofluorescence assay (IFA). Each compound was diluted to a final concentration of 100 nM and applied to BHK-21 or PK-15 cells infected with JEV or CSFV, respectively. Following 1 h of viral adsorption at 37 °C, cells were incubated in a maintenance medium containing the respective lipid compounds and harvested at 48 h (JEV) or 24 h (CSFV) post-infection. Cell viability assessed by the CCK-8 assay indicated that several compounds reduced viability in virus-infected groups, suggesting enhanced viral replication. Eight lipid compounds were identified that significantly increased JEV proliferation as determined by IFA (Figure 1A–E). The same compounds also markedly promoted CSFV replication in PK-15 cells compared with the DMSO control (Figure 1F,G). Further validation by RT-qPCR confirmed that glycerol phenylbutyrate, progesterone, hydrocortisone buteprate, protodioscin, heptadecanoic acid, 17-hydroxyprogesterone, allylestrenol, and BS significantly elevated total viral genome copy numbers in both JEV– and CSFV–infected cells. Among these, BS exerted the most pronounced effect, indicating a strong capacity to enhance CSFV replication.

### 3.2. BS Enhances CSFV Replication in a Dose-Dependent Manner

CCK-8 assays demonstrated that BS at increasing concentrations (100–500 nM) did not induce cytotoxicity (Figure 2A). Treatment of CSFV C-strain-infected cells with increasing concentrations of BS (100, 300, and 500 nM) resulted in a dose-dependent increase in viral RNA levels as quantified by RT-qPCR (Figure 2B). Western blotting further revealed that BS significantly upregulated CSFV E2 protein expression in both Shimen and C-strain infections (Figure 2C,D), indicating enhanced viral genome replication and protein synthesis. Collectively, these results demonstrate that BS facilitates CSFV replication in a concentration-dependent manner.

### 3.3. Differential Gene Expression Analysis (DEGs)

To investigate host transcriptional responses to BS during CSFV infection, RNA sequencing was performed on CSFV–infected PK-15 cells treated with or without BS. A total of 175 DEGs were identified, including 53 upregulated and 122 downregulated genes (Figure 3A). Hierarchical clustering of the six most upregulated and 14 most downregulated genes revealed significant enrichment in immune and inflammatory signaling pathways (Figure 3B). RT-qPCR validation of selected genes, including IκBα, ISG15, IL-8, and COX-2, confirmed their downregulation in BS–treated, CSFV–infected cells, consistent with transcriptomic results.

### 3.4. GO and KEGG Pathway Enrichment Analysis

Gene Ontology (GO) analysis indicated that DEGs were predominantly involved in biological regulation (172 genes), cellular processes (110), single-organism processes (105), and metabolic processes (77). In terms of molecular function, DEGs were enriched in categories such as binding (102 genes), catalytic activity (35), and signal transducer activity (14). Cellular component analysis revealed enrichment in the cell membrane (50 genes), organelles (46), and the extracellular region (21) (Figure 4A). KEGG pathway enrichment analysis revealed that DEGs were prominently associated with numerous immune and inflammatory pathways, including complement and coagulation cascades, IL-17, cytokine–cytokine receptor interaction, TNF, RIG-I-like receptor, p53, NF-κB, chemokine, NOD-like receptor, and PPAR pathways (Figure 4B). These findings suggest that BS modulates diverse host immune regulatory pathways during CSFV infection.

### 3.5. BS Downregulates IκBα and Enhances CSFV Infection

Among the most significantly downregulated genes identified by transcriptomic analysis, IκBα emerged as a key target affected by BS treatment. RT-qPCR analysis showed that both CSFV infection and BS treatment individually reduced IκBα mRNA levels, with a more pronounced reduction observed upon combined treatment (Figure 5A). Further experiments in CSFV–infected PK-15 cells treated with increasing concentrations of BS demonstrated a dose-dependent increase in viral RNA levels and p65 protein expression as measured by RT-qPCR and Western blotting. In parallel, both mRNA and protein levels of IκBα were significantly reduced (Figure 5B,C). These results suggest that BS enhances CSFV infection by downregulating IκBα, thereby potentially facilitating activation of the NF-κB signaling cascade.

### 3.6. Knockdown of IκBα Facilitates CSFV Replication

To investigate the role of IκBα in CSFV replication, PK-15 cells were transfected with either IκBα–specific small interfering RNA (siRNA) or non-targeting control siRNA. Western blotting analysis revealed that knockdown of IκBα significantly enhanced CSFV replication compared to the control group (Figure 6A). Subsequently, we infected IκBα–silenced cells with CSFV and treated them with BS (500 nM). RT-qPCR and Western blotting analyses showed that while IκBα knockdown alone promoted viral replication, the combined treatment with BS did not result in further enhancement compared to IκBα knockdown alone. These findings indicate that suppression of endogenous IκBα facilitates CSFV replication, and that the facilitated effect of BS may be mediated through downregulation of IκBα.

## 4. Discussion

BS, a plant-derived phytosterol, has garnered significant attention due to its diverse biological activities, including anti-inflammatory, antioxidant, and anticancer effects. Computational prediction of drug–target interactions has indicated a strong antiviral potential for BS, supported by its high binding affinity to the SARS-CoV-2 spike glycoprotein and the ACE2 receptor [16]. Interestingly, in plant–pathogen interactions, stigmasterol is often converted into BS in response to the microbial challenge, a process associated with increased pathogen proliferation and heightened plant susceptibility [17]. These findings suggest that elucidating the mechanisms by which BS modulates viral infections may facilitate the development of novel lipid-based therapeutic strategies.

This study demonstrated that BS significantly promotes the replication of both the Shimen and C strains of CSFV. Due to the absence of CPE during CSFV infection, direct compound screening remains challenging. JEV, a member of the same Flaviviridae family, shares similar replication mechanisms with CSFV and induces visible CPE in vitro. A dual-approach screening method combining indirect IFA and the CCK-8 cell viability assay was applied to test 135 lipid-related compounds. Eight compounds were identified that enhanced the replication of both CSFV and JEV: glycerol phenylbutyrate, progesterone, hydrocortisone buteprate, protodioscin, heptadecanoic acid, 17-hydroxyprogesterone, allylestrenol, and BS. Among these, BS exhibited the most potent pro-viral effect (*p* < 0.01), increasing the replication of JEV, CSFV-Shimen, and CSFV-C strains (*p* < 0.01). These results indicate that BS exerts a broad-spectrum effect on viral proliferation and represents a promising candidate for further mechanistic investigation. The CSFV C-strain is a widely used live-attenuated vaccine that offers a strong safety profile and induces both humoral and cellular immune responses in domestic and wild pigs. However, the immune-stimulating capacity of the C-strain does not fully replicate that of virulent strains. In immunocompromised animals or under suboptimal conditions, the vaccine may fail to elicit effective immunity [18]. Enhancing immune efficacy, reducing antigen doses, and lowering vaccine costs are essential goals often pursued through adjuvant optimization. The current study demonstrated that BS significantly and dose-dependently enhances replication of the CSFV C-strain. Supporting these findings, Fraile et al. reported that BS increased the proliferation of peripheral blood mononuclear cells (PBMCs) and activated porcine dendritic cells in vitro. In vivo, administration of phytosterols before PRRSV-MLV vaccination altered post-vaccination immune responses in pigs, including enhancing PBMC proliferation and elevating plasma apolipoprotein A1 levels, indicating that phytosterols may augment vaccine-induced immune responses [19].

To investigate the molecular mechanisms underlying BS-induced replication of CSFV, transcriptome profiling (RNA-Seq) was performed on CSFV–infected PK-15 cells treated with or without BS. A total of 228 differentially expressed genes (DEGs) were identified. Validation by RT-qPCR confirmed the downregulation of four representative genes, including IκBα, ISG15, IL-8, and COX-2, in agreement with RNA-Seq results. Consistent with these observations, BS has previously been shown to reduce the expression of influenza A virus (IAV)-induced proinflammatory mediators, including IL-6, TNF-α, IL-8, and COX-2(8). Production of COX-2 is regulated by NF-κB and p38 MAPK signaling and plays a pathogenic role during IAV infection [20,21]. GO and KEGG enrichment analyses revealed that BS downregulated key immune-related genes, such as PDE4B, DUOX2, IκBα, ISG15, and CXCL8, which are associated with innate immune responses and inflammation. These findings suggest that BS facilitates CSFV replication by modulating host immune and inflammatory signaling pathways in infected PK-15 cells. Previous studies have demonstrated that CSFV employs various immune evasion strategies, including suppression of PRR signaling, inhibition of IFN and NF-κB activation, lymphocyte depletion, and induction of autophagy, inflammation, and pyroptosis. These mechanisms act in concert to establish persistent infection and promote viral replication. The transcriptomic data presented here align with these known immune evasion mechanisms employed by CSFV [22].

Conversely, other reports have indicated that BS exhibits antiviral activity against IAV. It has been shown to inhibit IAV replication by targeting viral neuraminidase and M2 proteins, underscoring its potential as a broad-spectrum antiviral agent [23]. In vivo studies further demonstrate that BS alleviates IAV–induced proinflammatory responses and mitigates acute lung injury in mice, suggesting an immunomodulatory role during viral infection [8]. These contradictory findings, enhancement or inhibition of viral replication, may reflect context-dependent virus–host interactions. In the case of CSFV, BS may promote persistent infection by suppressing innate immune signaling, thereby facilitating viral replication.

NF-κB inhibitory protein (IκB) is an important member of the NF-κB signaling pathway, mainly regulating NF-κB activation and transcription. IκBα serves as a key modulator of both innate and adaptive immune responses across eukaryotic systems and is implicated in the development of various pathological conditions, including infectious diseases, immune dysfunctions, cancers, and genetic disorders. In the resting state, IκBα binds to the p65/p50 NF-κB heterodimer and prevents its nuclear translocation. In the classical NF-κB activation pathway, external stimuli trigger IKK–mediated phosphorylation of IκBα, leading to its ubiquitin-dependent degradation and subsequent release of the NF-κB complex, which then translocates into the nucleus to initiate gene transcription. Previous studies have shown that indoleamine 2,3-dioxygenase 1 (IDO1) facilitates CSFV replication by negatively regulating NF-κB signaling through tryptophan metabolism [24]. In contrast, curcumin has been reported to inhibit CSFV replication by enhancing innate immune responses without affecting NF-κB activation [15]. Both in vitro and in vivo analyses have demonstrated that CSFV infection does not significantly alter the NF-κB signaling pathway, suggesting that the virus may actively evade NF-κB–mediated immune responses during infection [25]. In this study, we found that β-sitosterol (BS) promotes the replication of classical swine fever virus (CSFV), accompanied by the downregulation of IκBα and enhanced activation of the NF-κB subunit p65. Moreover, silencing IκBα alone significantly enhanced CSFV proliferation, but BS treatment did not further enhance CSFV replication when IκBα was already silenced, suggesting that BS may promote viral replication by modulating IκBα protein expression and thereby mediating the NF-κB signaling pathway. Previous studies have also reported that the nonstructural protein NS2 of CSFV can activate the NF-κB transcription factor, thereby inducing endoplasmic reticulum (ER) stress in porcine umbilical vein endothelial cells. This activation promotes the expression of interleukin-8 (IL-8) and the anti-apoptotic protein Bcl-2, ultimately conferring resistance to proteasome inhibitor MG132–induced apoptosis [26]. These findings raise the possibility that CSFV may exploit multiple, potentially coordinated, mechanisms involving NF-κB modulation, the precise contributions of which warrant further investigation.

## 5. Conclusions

Collectively, the findings indicate that BS promotes CSFV replication by modulating host immune pathways and alleviating virus-induced immune pressure. The dual immunomodulatory properties of BS suggest potential applications both as an immunopotentiator to enhance vaccine efficacy and as a molecular target for the development of antiviral therapies.

## Figures and Tables

**Figure 1 viruses-17-00933-f001:**
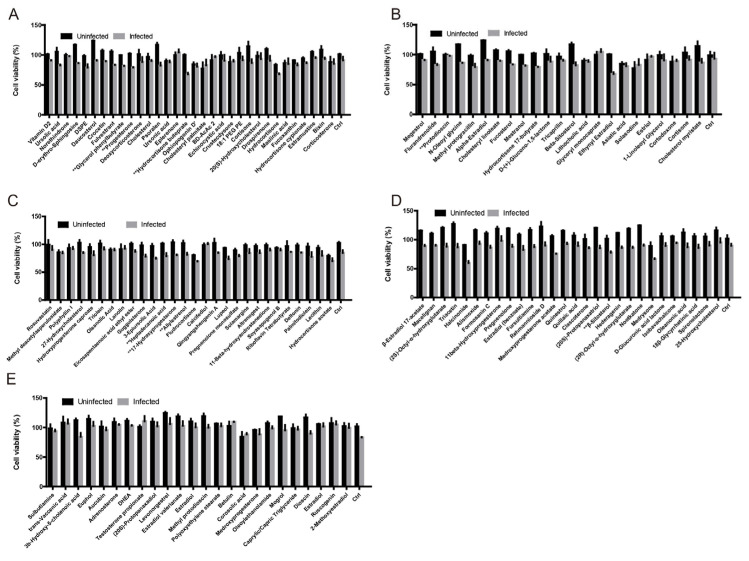
Screening of lipid compounds that promote CSFV proliferation. (**A**–**E**) Cells were infected with JEV (MOI = 0.5) for 48 h in the presence of 135 lipid compounds. Cell viability was assessed using the CCK-8 assay. (**F**) PK-15 cells were infected with CSFV (MOI = 1) and treated with DMSO or selected lipid compounds. Infected cells were stained for viral proteins (green) and nuclei (DAPI, blue), and visualized by fluorescence microscopy. (**G**) The percentage of CSFV–infected cells was quantified using ImageJ software. (**H**) PK-15 cells treated with the top eight candidate compounds (100 nM) were infected with CSFV or JEV (MOI = 0.5) for 24 h. Total viral RNA levels were quantified by RT-qPCR. Data are shown as mean ± SD from three independent experiments. * *p* < 0.05; ** *p* < 0.01.

**Figure 2 viruses-17-00933-f002:**
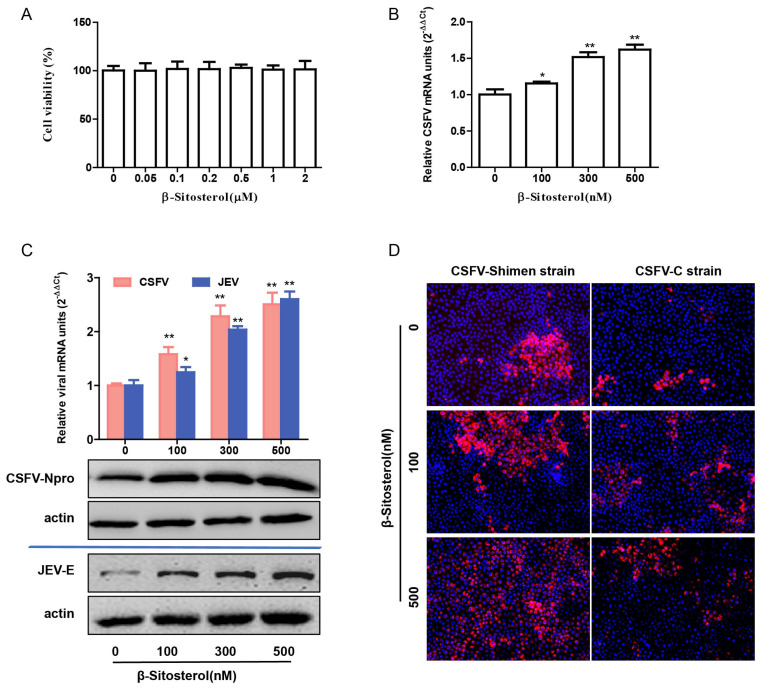
BS enhances CSFV replication in a dose-dependent manner. (**A**) Cytotoxicity of BS was evaluated using the CCK-8 assay. (**B**) PK-15 cells were treated with DMSO or BS (100, 300, or 500 nM) and infected with the CSFV-C strain (MOI = 1) for 24 h. Viral RNA was quantified by RT-qPCR. (**C**) Similar treatment was conducted using the CSFV-Shimen strain, followed by RT-qPCR and Western blotting. (**D**) Cells infected with CSFV-C or CSFV-Shimen (MOI = 1) were treated with different concentrations of BS. Viral proteins (red) and nuclei (DAPI, blue) were visualized under a fluorescence microscope. Data are shown as mean ± SD from three independent experiments. * *p* < 0.05; ** *p* < 0.01.

**Figure 3 viruses-17-00933-f003:**
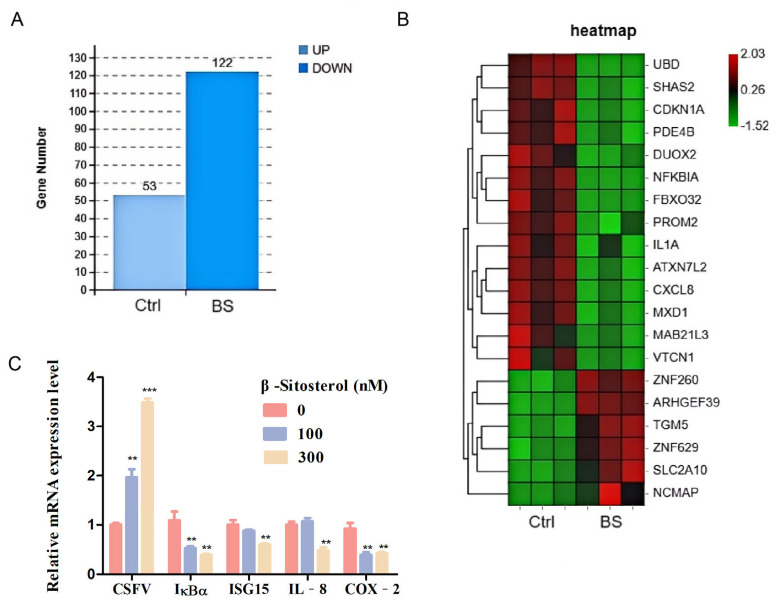
Transcriptomic analysis of PK-15 cells infected with CSFV and treated with BS. (**A**) PK-15 cells were pretreated with 100 nM BS or DMSO for 1 h, followed by CSFV infection (MOI = 1) for 24 h. RNA-Seq identified 175 DEGs, including 53 upregulated and 122 downregulated genes in the BS group. (**B**) Heatmap of DEGs; red indicates upregulation, and green indicates downregulation. (**C**) RT-qPCR validation of selected immune-related genes (IκBα, ISG15, IL-8, and COX-2) in BS–treated, CSFV–infected cells. Data are shown as mean ± SD from three independent experiments. ** *p* < 0.01, *** *p* < 0.001.

**Figure 4 viruses-17-00933-f004:**
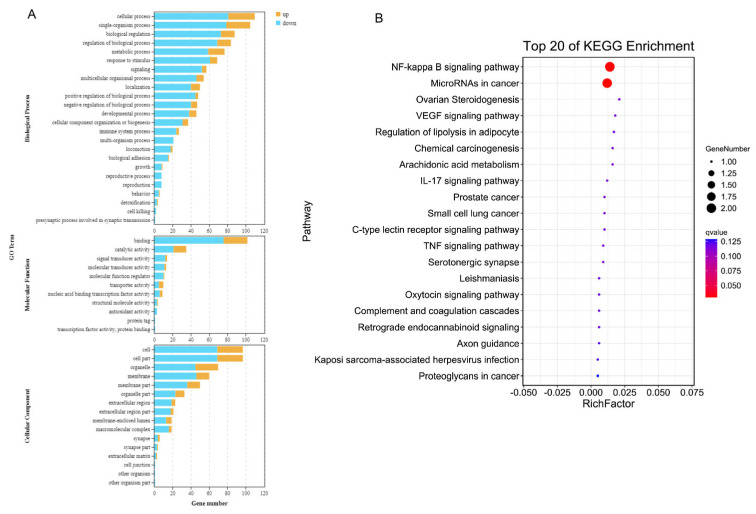
GO and KEGG enrichment analyses of BS–induced DEGs in CSFV–infected cells. (**A**) GO enrichment analysis categorized DEGs into Biological Process (BP), Molecular Function (MF), and Cellular Component (CC). (**B**) Bubble plot showing the top 15 significantly enriched KEGG pathways.

**Figure 5 viruses-17-00933-f005:**
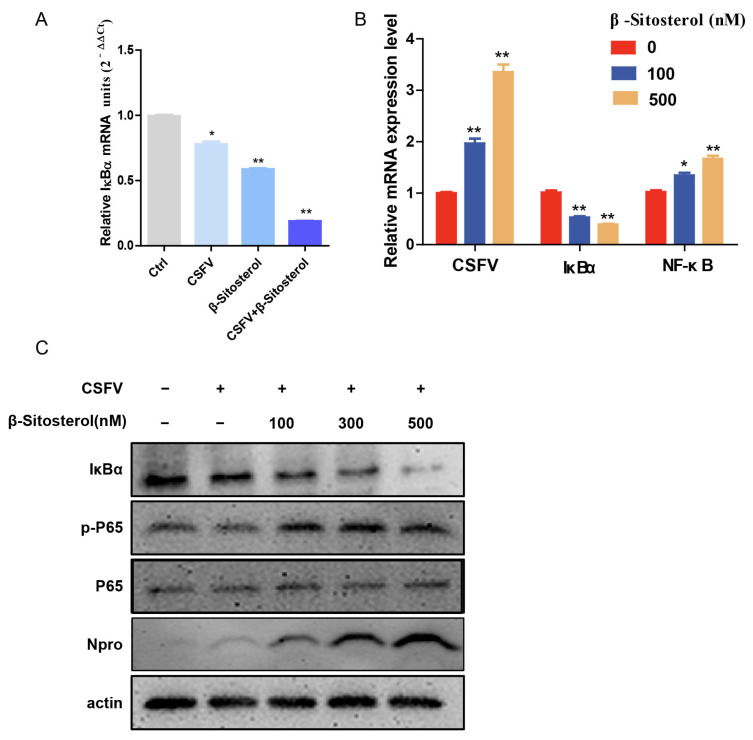
BS promotes CSFV replication by downregulating IκBα expression. (**A**) PK-15 cells were treated with 100 nM BS and infected or not with CSFV (MOI = 1) for 24 h. Viral RNA levels were measured by RT-qPCR. (**B**) Cells were treated with increasing concentrations of BS (0, 100, and 500 nM), followed by CSFV infection (MOI = 1) for 24 h. RT-qPCR was performed to quantify viral RNA and host NF-κB/IκBα transcripts. (**C**) Western blotting analysis of NF-κB pathway-related proteins (IκBα, p-P65, and P65) and viral protein Npro. β-actin served as the loading control. Data are shown as mean ± SD from three independent experiments. * *p* < 0.05; ** *p* < 0.01.

**Figure 6 viruses-17-00933-f006:**
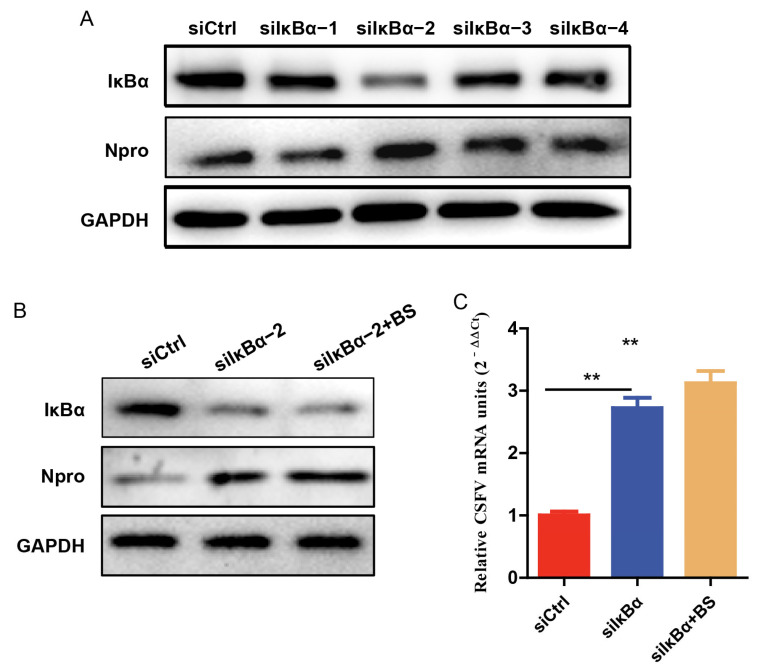
Silencing of IκBα enhances CSFV replication. (**A**) Cells were transfected with either siCtrl or siIκBα for 24 h before infection with CSFV (MOI of 0.1); samples were collected 24 h post-infection for Western blotting to determine the knockdown efficiency of IκBα and evaluate viral protein expression. (**B**,**C**) Cells were transfected with either siCtrl or siIκBα for 24 h before infection with CSFV (MOI of 0.1), and treated with BS (500 nM) for 24 h. Western blotting was performed to detect the protein levels of IκBα and CSFV Npro, and RT-qPCR was conducted to quantify total viral RNA. β-actin was used as a loading control. Data are shown as mean ± SD from three independent experiments. ** *p* < 0.01.

## Data Availability

The data used and/or analysed during the current study are available from the corresponding author upon reasonable request.

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
