# Peer review of "Beta-Sitosterol Enhances Classical Swine Fever Virus Infection: Insights from RNA-Seq Analysis"

_viruses, 2025, doi:10.3390/v17070933_

Round 1
Reviewer 1 Report
Comments and Suggestions for Authors
The manuscript entitled “Beta-sitosterol promotes classical swine fever virus replication by modulating host immune pathways” presents a well-conducted and scientifically relevant study that contributes significantly to the understanding of lipid-host interactions in classical swine fever virus (CSFV) infection. The authors have successfully demonstrated that beta-sitosterol (BS), a plant-derived phytosterol, enhances CSFV replication in PK-15 cells through the modulation of host immune pathways, particularly via downregulation of IκBα and activation of the NF-κB signaling cascade.
The experimental design is robust, with clear objectives and appropriate use of methodologies such as immunofluorescence assays, cell viability assessments, RNA sequencing, and RT-qPCR validation. The screening of 135 lipid compounds for their effects on viral replication is methodologically sound, and the identification of BS as a potent enhancer of both virulent and attenuated CSFV strains is a valuable contribution to the field. The transcriptomic analysis further supports these findings by revealing key differentially expressed genes involved in immune and inflammatory responses, providing a molecular basis for the observed effects.
The discussion effectively contextualizes the findings within the broader scope of antiviral research and highlights the potential applications of BS as an adjuvant to enhance vaccine efficacy or as a target for novel antiviral therapies. The manuscript is well-written, logically structured, and supported by comprehensive data that strengthen the conclusions.
A few minor points could be addressed to further improve the clarity and impact of the manuscript:
1. While the study provides strong evidence that BS modulates host immune pathways to promote CSFV replication, a more detailed explanation of the direct interaction between BS and specific host or viral proteins would add depth to the mechanistic interpretation.
2. A brief clarification regarding the rationale for using JEV as a surrogate model for CSFV screening would help readers better understand the experimental approach.
Overall, this manuscript makes a solid and timely contribution to the field of virology and antiviral drug development. With only minor revisions to address the above points, the paper is suitable for publication and will likely be of interest to researchers in the field.
Author Response
Comments 1: While the study provides strong evidence that BS modulates host immune pathways to promote CSFV replication, a more detailed explanation of the direct interaction between BS and specific host or viral proteins would add depth to the mechanistic interpretation.
Response 1: Thank you for your comments. Previous studies have reported that CSFV infection for up to 36 h has no significant effect on the NF-κB signaling pathway[1]. However, upon β-sitosterol (BS) treatment, differential gene expression analysis identified 53 upregulated and 122 downregulated genes, among which NF-κB pathway components were notably altered. Both Western blotting and RT-qPCR results demonstrated activation of the NF-κB signaling pathway following BS treatment. To further validate the involvement of this pathway, we performed siRNA-mediated knockdown of IκBα in PK-15 cells, followed by BS treatment. The results showed that knockdown of IκBα alone significantly promoted CSFV replication. However, the combined treatment with BS in IκBα-silenced cells did not lead to a further increase in viral replication (Fig. 6). These findings suggest that BS may promote viral proliferation by modulating the IκBα. However, the specific mechanism by which BS interacts with IκBα and CSFV proteins to regulate viral replication remains to be elucidated and will be investigated in future studies.
Comments 2: A brief clarification regarding the rationale for using JEV as a surrogate model for CSFV screening would help readers better understand the experimental approach2.
Response 2: Thank you for the constructive comments. First, both Japanese Encephalitis Virus (JEV) and Classical Swine Fever Virus (CSFV) belong to the *Flaviviridae* family and share similar pathogenic mechanisms. Second, JEV infection typically induces obvious cytopathic effects (CPE) and cell damage, resulting in a measurable decrease in cell viability. Therefore, we used a cell viability assay (CCK-8) to assess the effect of 135 lipid compounds on JEV replication. In contrast, CSFV is known to suppress host immune responses during infection, enabling it to establish persistent infection without inducing significant cytopathic effects in the short term. As a result, changes in cell viability are minimal during early-stage CSFV infection, and indirect immunofluorescence assay (IFA) is required to monitor the level of viral replication. In summary, using complementary screening approaches, we identified eight lipid compounds that promote the replication of both JEV and CSFV. These findings may provide new theoretical support for the discovery of host-directed antiviral targets against flaviviruses.
- Gao, Y.; Hu, J. H.; Liang, X. D.; Chen, J.; Liu, C. C.; Liu, Y. Y.; Cheng, Y.; Go, Y. Y.; Zhou, B., Curcumin inhibits classical swine fever virus replication by interfering with lipid metabolism. Vet Microbiol 2021,259, 109152. 10.1016/j.vetmic.2021.109152
Reviewer 2 Report
Comments and Suggestions for Authors
This is an in vitro experiment showing that beta-sitosterol can suppress IκBα expression thereby result in enhancing the CSFV replication. There is not much benchwork and most of the output are by software analysis.
The authors start with a lipid compound library (HY-L043, MedChemExpress, USA). What authors did not reveal is how do they identify this beta-sitosterol out of the whole library.
The first experiment identify 53 upregulated and 122 downregulated genes, and authors select a few of those on top of both categories to further validate by RT-qPCR and luckily found that IκBα is one of those key target of this phytocompound.
General comment: IκBα is a pathway that many biological activities employed, and since there are another 121 genes being downregulated by beta-sitosterol, one cannot rule out the effect is totally contributed by best-sitosterol, or vice versa via the IκBα. Unless authors employed PK15 cells with IκBα knockout or with purified beta-sitosterol, all these claims are only suggestive.
Author Response
Comments 1: The authors start with a lipid compound library (HY-L043, MedChemExpress, USA). What authors did not reveal is how do they identify this beta-sitosterol out of the whole library.
Response 1: Thank you for the constructive comments. In the revised manuscript (Results, Section 3.1. Lipid Compounds Promote CSFV Proliferation, p.3-p.4, ), we have clarified that beta-sitosterol (BS) was identified based on its consistent enhancement of viral replication across both JEV and CSFV systems, as measured by CCK-8 viability assays, immunofluorescence, and RT-qPCR. It was selected as the top candidate among eight lipid compounds that showed significant enhancement of viral RNA levels.
Comments 2: The first experiment identify 53 upregulated and 122 downregulated genes, and authors select a few of those on top of both categories to further validate by RT-qPCR and luckily found that IκBα is one of those key target of this phytocompoundGeneral comment: IκBα is a pathway that many biological activities employed, and since there are another 121 genes being downregulated by beta-sitosterol, one cannot rule out the effect is totally contributed by best-sitosterol, or vice versa via the IκBα. Unless authors employed PK15 cells with IκBα knockout or with purified beta-sitosterol, all these claims are only suggestive.
Response 2: Thank you for the constructive comments. Previous studies have shown that CSFV infection for up to 36 h has no detectable effect on the NF-κB signaling pathway. However, upon treatment with β-sitosterol (BS), differential gene expression analysis identified the NF-κB pathway were notably affected. Consistently, both Western blotting and RT-qPCR results demonstrated that BS treatment led to activation of the NF-κB pathway. To further validate the involvement of this pathway, we performed siRNA-mediated knockdown of IκBα followed by BS treatment. The results showed that silencing IκBα alone significantly promoted CSFV replication, but the addition of BS to IκBα-deficient cells did not result in a further increase in viral replication (Fig. 6). These findings suggest that BS may promote viral proliferation primarily by suppressing IκBα, and that BS's effect on promoting viral proliferation reaches saturation after IκBα is silenced.
Reviewer 3 Report
Comments and Suggestions for Authors
In this manuscript “Beta-Sitosterol Enhances Classical Swine Fever Virus Infection: Insights from RNA-Seq Analysis” by Liu et al., the authors have studied effect of beta-sitosterol in classical swine fever virus (CSFV) replication and virus induced host innate immune response. The study shows beta-sitosterol enhances CSFV replication and regulates NFkB pathway. The study well written, with only few minor comments
Specific comments:
- Abstract section: P.No.1, Line No. 24-27. This is over speculation. Since the study is based on in vitro model, it is unwarranted to speculate on role of beta-sitosterol in increase CSFV vaccine immunogenicity. Please remove this line.
- Results section: P.No.5, Line No. 146. Please mention extracellular or intracellular viral RNA load in this section and in the corresponding figure legend.
- Discussion section. P.No.10, Line No. 296-297. Please remove this statement, it is overspeculation. The current study and the Ref# 26, estimates virus load at 24hpi. Therefore, it is impossible to speculate on CSFV persistence without experiments to delineate virus load at various time points, both in vitro and in vivo model. Virus persistence is usually chronic, latent, and/or silent – virus shedding / circulation occurring for longer period of time.

Author Response
Comments 1: Abstract section: P.No.1, Line No. 24-27. This is over speculation. Since the study is based on in vitro model, it is unwarranted to speculate on role of beta-sitosterol in increase CSFV vaccine immunogenicity. Please remove this line.1.
Response 1: Thank you for your suggestion. We agree and have removed the speculative sentence regarding enhanced vaccine immunogenicity from the Abstract.
Comments 2: Results section: P.No.5, Line No. 146. Please mention extracellular or intracellular viral RNA load in this section and in the corresponding figure legend.
Response 2: Thank you for this clarification request. We have updated the Results section ( P.No.4, Line No. 142.) and the corresponding figure legend (P.No.5, Line No. 152) to indicate that the measured viral RNA represents total RNA which including extracellular and intracellular viral load. The experimental procedure was as follows: PK-15 cells were seeded into 24-well plates, infected with CSFV, and treated with eight different lipid compounds. After 24 hours, the samples were collected and subjected to two freeze-thaw cycles. Total RNA was then extracted using TRIzol reagent. The RNA copy numbers of CSFV and cellular factors were quantified by RT-qPCR using the 7300 Real-Time PCR System.
Comments 2: Discussion section. P.No.10, Line No. 296-297. Please remove this statement, it is overspeculation. The current study and the Ref# 26, estimates virus load at 24hpi. Therefore, it is impossible to speculate on CSFV persistence without experiments to delineate virus load at various time points, both in vitro and in vivo model. Virus persistence is usually chronic, latent, and/or silent – virus shedding / circulation occurring for longer period of time.
Response 3: Thank you for your comments. We appreciate this valuable comment and agree that our current data do not support conclusions about viral persistence. We have removed the speculative sentence from the Discussion section. Additionally, reference 26 has been replaced to make the manuscript regulations clearer.
Round 2
Reviewer 2 Report
Comments and Suggestions for Authors
Authors have responded to my two comments and the key information resided in figure 1 and figure 6.
It is now clear that this compound BS is demonstrated effective and feasible in PK-15 cells, there is still a long road to show it useful in animal body.
reference 16 and 27 sources are incomplete.
Author Response
Comments 1: reference 16 and 27 sources are incomplete.
Response 1: Thank you for your comments. Regarding the reference issue, we apologize for the oversight. References 16 and 27 have now been carefully checked and updated with complete citation details in the revised manuscript.